# A Novel P/N/Si-Containing Vanillin-Based Compound for a Flame-Retardant, Tough Yet Strong Epoxy Thermoset

**DOI:** 10.3390/polym15102384

**Published:** 2023-05-19

**Authors:** Siyuan He, Cheng Chi, Chaohua Peng, Birong Zeng, Yongming Chen, Zhongxi Miao, Hui Xu, Weiang Luo, Guorong Chen, Zhenping Fu, Lizong Dai

**Affiliations:** 1Fujian Provincial Key Laboratory of Fire Retardant Materials, College of Materials, Xiamen University, Xiamen 361005, China; 20720201150052@stu.xmu.edu.cn (S.H.); 20720201150039@stu.xmu.edu.cn (C.C.); 20720200155774@stu.xmu.edu.cn (C.P.); chenym971213@163.com (Y.C.); 20720201150122@stu.xmu.edu.cn (Z.M.); 20720201150095@stu.xmu.edu.cn (H.X.); luoweiang@xmu.edu.cn (W.L.); grchen@xmu.edu.cn (G.C.); 20720221150110@stu.xmu.edu.cn (Z.F.); 2Xiamen Key Laboratory of Fire Retardant Materials, College of Materials, Xiamen University, Xiamen 361005, China

**Keywords:** flame retardant, vanillin, epoxy resin, synergistic effect, strength and toughness

## Abstract

It is still extremely challenging to endow epoxy resins (EPs) with excellent flame retardancy and high toughness. In this work, we propose a facile strategy of combining rigid–flexible groups, promoting groups and polar phosphorus groups with the vanillin compound, which implements a dual functional modification for EPs. With only 0.22% phosphorus loading, the modified EPs obtain a limiting oxygen index (LOI) value of 31.5% and reach V-0 grade in UL-94 vertical burning tests. Particularly, the introduction of P/N/Si-containing vanillin-based flame retardant (DPBSi) improves the mechanical properties of EPs, including toughness and strength. Compared with EPs, the storage modulus and impact strength of EP composites can increase by 61.1% and 240%, respectively. Therefore, this work introduces a novel molecular design strategy for constructing an epoxy system with high-efficiency fire safety and excellent mechanical properties, giving it immense potential for broadening the application fields of EPs.

## 1. Introduction

As one of the essential thermosetting polymers, epoxy resins (EPs) are widely utilized in various industrial fields owing to their outstanding adhesion, chemical resistance, and electrical insulation [1,2]. EPs refer to a type of polymer containing two or more epoxy groups, which can form a curing cross-linking network under the action of compounds containing active hydrogen [3,4]. Diglycidyl ether of bisphenol-A (DGEBA) as an epoxide compound prepared from epichlorohydrin and bisphenol A is the most widely used type of EP, accounting for approximately 90% of the world’s total production of EPs [5]. Unfortunately, the high release of toxic flue gas upon combustion, flammability, and high brittleness greatly limit its further applications in advanced manufacturing fields such as chip encapsulation and smart devices [6,7]. The low impact strength and high brittleness of EPs will tremendously increase the odds of mechanical damage to EP-based electronics, thereby limiting their service life. Nanomaterials such as graphene [8], boron nitride [9], and zinc oxide [10] produce surface and interface effects with the EP matrix, which can improve the impact resistance of EPs and dissipate the energy caused by bending or impact, but nanomaterials suffer from poor dispersity and a tedious preparation process [11]. Therefore, exploring novel multi-functional molecules to elevate fire safety and boost the mechanical strength of EPs is regarded as a facile and effective approach.

In recent years, people have paid more attention to carbon emissions and environmental pollution issues in the preparation of polymers [12]. Emerging as the times require, renewable bio-based materials such as vanillin [13,14], phytic acid [15], cardanol [16], and furfural [17] have gradually become a new choice for the design of flame retardants thanks to their multiple reaction sites and renewable advantages. Among these raw materials, vanillin has active phenolic hydroxyl and aldehyde groups, making it a favorite in industrial production and an ideal choice for building multifunctional bio-based flame retardants [13]. Regrettably, flame retardants prepared entirely from bio-based compounds have limited effectiveness [18]. Therefore, it is necessary to introduce efficient flame-retardant elements. Halogen-containing flame retardants have high flame retardancy efficiency but produce hypertoxic substances, which do not conform to environmentally friendly needs [19]. In other words, combining safe non-halogen elements (e.g., phosphorus [20], nitrogen [21], silicon [22], or transition metals [23]) with bio-based materials is a potential strategy for preparing bio-based flame retardants. For instance, Zhu et al. [24] successfully synthesize a eugenol-based flame retardant, EGN-Si/P. Compared with pure EPs, the EP composite containing 5 wt% EGN-Si/P can achieve the UL-94 V0 grade and improve its impact strength by 45.6%, although its tensile strength and glass transition temperature (T_g_) decrease. Wang et al. [25] use the neutralization reaction between phytic acid and imidazole to synthesize a biological-based flame retardant, IMPA. Although the imidazole groups promote the curing process, the EP/IMPA composite requires up to 15% IMPA addition to pass the UL-94 V0 vertical burning test and reduces the storage modulus. Thereby, introducing curing-promoting imidazole groups and flexible siloxane structures into the design of flame retardants may help improve flame retardancy and potentially enhance the strength and toughness of EPs. Unfortunately, there are few reports on related work.

P-containing organics are favored for their outstanding flame-retardant performance in both gas and condensed phases [26]. 9,10-dihydro-9-oxo-10-phosphophenanthrene-10-oxide (DOPO) and its derivatives have become one of the representatives of organophosphorus flame retardants because of their high activity, wide applicability, and remarkable effectiveness [27,28]. Accordingly, DOPO is an ideal candidate for designing bio-based flame retardants. Nevertheless, flame retardants containing a single component of phosphorus need more additions to achieve the flame-retardant effect, which will weaken the thermal properties of EP, such as the reduction in glass transition temperature (T_g_) and the initial decomposition temperature (T_5%_) [29]. In other words, reducing the P content in the design of flame retardants will reduce the negative impact of unstable phosphate groups on the thermal stability of materials. P/N-containing flame retardants can help EPs form carbon layers catalyzed by phosphoric acid and release a large amount of nitrogen-containing gas during combustion, showing a highly efficient flame retardancy. Tang et al. [30] synthesize a flame retardant (TAD) containing phosphaphenanthrene and triazine groups. When the mass fraction of TAD reaches 12 wt%, the EP thermoset acquires the highest LOI value of 33.5% and the UL94 V-0 rating. Nabipour et al. [31] successfully prepare a vanillin-derived cyclotriphosphazene-cored triazole compound (HHCTP). The EP thermoset containing 7.5 wt% HHCTP achieved a UL-94 V-0 rating and an LOI value of 31.5%. Besides, the EP/HHCTP composites maintain excellent thermomechanical properties. Considering these factors, this work aims to minimize the introduction of phosphorus as much as possible and maximize flame retardancy by combining P-containing DOPO with N-containing benzimidazole.

Overall, the purpose of this work is to construct an EP system with high fire safety, high strength, and high toughness. A novel P/N/Si-containing vanillin derivative (DPBSi) is easily prepared by the Kabachnik–Fields reaction and applied as the flame retardant and reinforcing agent for EP. The successful preparation of DPBSi is proved using NMR and FTIR. More essentially, the experimental results indicate that the synergistic flame-retardant effect of P/N/Si elements improves the fire safety and smoke suppression of EP. When the P content is only 0.22 wt%, the modified EP can reach the V0 grade in UL-94 tests. Profiting from the curing-promoting effect of the benzimidazole groups and the flexible phenylsiloxane structure, the mechanical properties are strengthened. Besides, the influencing mechanism of DPBSi on mechanical properties and thermal decomposition behaviors of EP are studied in detail. This work is expected to provide brand new ideas for the design of multifunctional bio-based flame retardants using vanillin as a building block and the preparation of a high-performance epoxy thermoset.

## 2. Experimental Section

### 2.1. Materials

2-aminobenzimidazole (ABZ), diphenyldichlorosilane, vanillin, 4,4′-diaminodiphenylme-thane (DDM), and 9,10-dihydro-9-oxo-10-phosphophenanthrene-10-oxide (DOPO) were purchased from Shanghai Aladdin Biochemical Technology Co. Ltd., Shanghai, China. Ethanol, methylene chloride, 1,4-dioxane, anhydrous magnesium sulfate, and triethylamine were bought from Xilong Chemical Co. Ltd., Shantou, China. Diglycidyl ether of bisphenol-A (DGEBA) with an epoxy value of 0.51 was provided by Lanxing resin factory, Wuxi, China.

### 2.2. Synthesis of DPBSi

According to the literature, VSi-CHO is synthesized [32]. DPBSi is synthesized simply by the “one-pot” process. Firstly, 7.456 g (0.056 mol) 2-Aminobenzimidazole and 12.6 g (0.026 mol) VSi-CHO are added into a triple flask containing 50 mL 1, 4-dioxane. Under the protection of nitrogen, the reaction is stirred at 75 °C for 8 h. When the reaction is complete, 35 mL of 1, 4-dioxane solution dissolved with 12.1 g (0.056 mol) DOPO is added to the reactor by dropping. Afterward, the reaction is continued at reflux temperature for 18 h. In the end, the white product is separated by filtration, washed with absolute ethanol two to three times, and dried in a 60 °C vacuum oven for 24 h (yield: 92%). The synthesis route is exhibited in Figure 1.

### 2.3. Preparation of DGEBA/DDM/DPBSi Mixtures for DSC Tests

According to the formulations shown in Table 1, DGEBA and DPBSi are ultrasonically dispersed in 150 mL acetone. Then, the DGEBA/DPBSi mixture is obtained by removing acetone using vacuum distillation at 50 °C. The mixtures of DGEBA/DDM/DPBSi and DGEBA/DDM are also obtained using the above method. All samples are used for the differential scanning calorimeter (DCS) test.

### 2.4. Synthesis of the Cured Epoxy Resins

At first, different qualities (see Table 1) of DPBSi and DGEBA are stirred at 150 °C for 40 min with a magnetic stirrer. After uniform mixing, water and air are removed by a vacuum pump. When the reactor’s temperature cools to 80 °C, the curing agent (DDM, 25 g) is added and the stirring of the mixture continues for 20 min. Then, the liquid mixture is quickly decanted into the preheated mold and placed into the air-circulating oven. The curing procedure is to hold the temperature at 130 °C, 160 °C, and 180 °C for 2 h, respectively. Similar curing procedures are used to prepare EP without DPBSi. The corresponding formulations are illustrated in Table 1.

### 2.5. Characterization

Fourier transform infrared (FTIR) spectra were conducted on Nicolet iS10 Spectrometer (Thermo Fisher, Waltham, WA, USA), using ATR mode, in the wavenumber range of 4000–525 cm^−1^.

^1^H, ^31^P, and ^13^C nuclear magnetic resonance (NMR) spectra were analyzed using an Advanced III HD 500 MHz spectrometer (Bruker, Fällanden, Switzerland) and DMSO-d6 as solvent.

Curing curves of EP and other EP composites were carried out using a differential scanning calorimeter (DSC25, TA Instruments, New Castle, DE, USA) under the N_2_ atmosphere. Each liquid sample (about 6–10 mg) was heated from 25 °C to 270 °C at a rate of 5 °C/min.

Thermogravimetric analysis (TGA) was conducted on a Netzsch STA 409EP thermogravimetric analyzer (Netzsch, Selb, Germany) under the N_2_ and air atmosphere. Each powder sample (about 6–10 mg) was heated from 25 °C to 800 °C at a rate of 10 °C/min.

Dynamic thermomechanical analysis (DMA) was performed on a Netzsch DMA 242E (Netzsch, Selb, Germany) instrument with a size of 60 × 10 × 3.3 mm^3^. Each sample bar was heated from 25 °C to 220 °C at a rate of 1 °C/min.

The hardness and Young’s modulus were studied by the depth-sensing indentation (DSI) using a CSM NHT2 instrument (CSM Instruments SA, Peseux, Switzerland), with a size of 60 × 10 × 4 mm^3^.

The three-point bending tests were studied using a CMT6503 Testing Machine (MTS Industrial Systems, Shenzhen, China), with a size of 60 × 10 × 4 mm^3^, referring to the GB/T 9341-2008.

The charpy notched impact strength was measured with a ZBC1400-B Pendulum Impact Testing Machine (MTS Industrial Systems, Shanghai, China), with a size of 80 × 10 × 4 mm^3^, referring to the GB/T 1043.1-2008.

Vertical burning tests (UL-94) were carried out using the FTT0082 combustion device (FTT, East Grinstead, UK), referring to the ASTM D3801, with a size of 125 × 13 × 3.5 mm^3^.

The limiting oxygen index (LOI) was obtained using a FTT0077 Oxygen Index Tester (FTT, East Grinstead, UK), according to the ASTM D2863, with a size of 100 × 6 × 3.5 mm^3^.

The cone calorimeter (CC) test was conducted on an FTT0484 machine (FTT, East Grinstead, UK) at 35 kW/m^2^ heat flux, referring to ISO 5660, with a size of 100 × 100 × 3.0 mm^3^. The weight of the test sample was guaranteed at 30 ± 0.6 g.

TG-FTIR results were collected on a PE STA 6000 Frontier instrument (Physical Electronics Inc., Chanhassen, MN, USA) under the N_2_ atmosphere. Each powder sample was heated from 25 °C to 850 °C at a rate of 10 °C/min.

Scanning electron microscopy (SEM, Su-70, Hitachi Hi-tech Nagase Office, Tokyo, Japan) was used to observe the microstructure of char residues and fractography morphologies of EPs and EP composites.

Raman spectra were conducted on an X-polar spectrometer (Horiba, Tokyo, Japan) and the excitation wavelength was 532 nm.

X-ray photoelectron spectroscopy (XPS) was performed on a Quantum2000 tester (Physical Electronics Inc, Chanhassen, MN, USA) with Al Kα (1486.6 eV) radiation.

## 3. Results and Discussion

### 3.1. Characterization

The FTIR spectra of ABZ, VSi-CHO, DOPO, and DPBSi are shown in Figure 1. Compared with VSi-CHO and DOPO, DPBSi does not present the characteristic peaks of -CHO at 2842 cm^−1^ (C-H), 2735 cm^−1^ (C-H), and 1690 cm^−1^ (C=O) [14]. As expected, the double peaks (3381 cm^−1^ and 3321 cm^−1^) representing -NH_2_ merge into the single peak (3328 cm^−1^) that corresponds to -NH. Furthermore, DPBSi retains the peak of C-H on the methoxy group at 2921 cm^−1^ and the peak of C=N on the benzimidazole ring at 1633 cm^−1^ [33]. The characteristic peak of P-O-Ph appears at 753 cm^−1^, but the characteristic peak of P-H at 2438 cm^−1^ disappears [34]. These results indicate that DOPO has completely reacted with Schiff base intermediates and DPBSi has been synthesized correctly.

To further prove the successful synthesis of DPBSi, the ^1^H NMR, ^31^P NMR, and ^13^C NMR spectra of DPBSi are presented in Figure 2. In the ^1^H NMR spectrum of VSi-CHO shown in Appendix A, the peak at 9.83 ppm corresponds with -CHO (Ha), the peaks at 6.95–8.00 ppm belong to aromatic hydrogen protons, and the peaks at 3.51–3.92 ppm are attributed to -OCH_3_ (Hb). In the ^1^H NMR spectrum of DPBSi, the single peak at 10.34 ppm is attributed to -NH (Ha) in the benzimidazole ring and the signal of -NH (Hb) is linked to the benzimidazole ring that appears at 9.03 ppm, while the peak of -CH arises at 5.32–5.40 ppm and the peaks of hydrogen protons on benzene rings appear at 6.58–8.42 ppm [35]. Compared with the ^1^H NMR spectra of VSi-CHO, the peaks of -CHO disappear, while there are similar peaks of -OCH_3_ (Hd) at 3.68–3.79 ppm in the ^1^H NMR spectrum of DPBSi. More importantly, the integral area ratio of hydrogen atoms is generally consistent with the theoretical ratio in VSi-CHO and DPBSi. By comparing the ^31^P NMR spectra in Figure 2b, it is distinct that the two products display different chemical shifts. DOPO shows two peaks at 14.51 ppm and 15.72 ppm, whereas DPBSi shows two phosphorus signals at 29.07 ppm and 31.29 ppm. As shown in Figure 2c, a total of 29 carbon signals in DPBSi are confirmed, including several key carbon signals (C_1_ = 56.03 ppm, C_2_ = 66.83 ppm, C_6_ = 149.56 ppm, C_7_ = 147.72 ppm, and C_16_ = 131.31 ppm).

Based upon the above analyses, the successful synthesis of DPBSi is proved.

### 3.2. Curing Behaviors and Thermal Stability

Differential scanning calorimetry (DSC) was applied to discuss whether or not DPBSi affects the curing process of the EP system. As shown in Figure 3, there is a wide exothermic peak (146.1 °C) in the DGEBA/DDM curve. Besides, the introduction of DPBSi does not have a distinct influence on the curing activity, and the exothermic peaks are 146.7 °C for DGEBA/DDM/DPBSi-2, 149.6 °C for DGEBA/DDM/DPBSi-4, and 145.7 °C for DGEBA/DDM/DPBSi-6, while the exothermic peak of DGEBA/DPBSi is up to 226.8 °C, which is attributed to the ring-opening of benzimidazole groups [36,37]. These results illustrate that N-H bonds on the benzimidazole groups are still active and do not participate in the curing process by chemical crosslinking.

Thermogravimetric analysis (TGA) and differential thermogravimetry analysis (DTG) are utilized to evaluate the thermal stability of EP, DPBSi, and EP/DPBSi composites under the N_2_ atmosphere. Figure 4 shows thermal decomposition curves and the relevant results are listed in Table 2.

In Figure 4b, DPBSi has two decomposition stages, the initial decomposition temperature (T_5%_) is 295.1 °C, and the residual mass is 31.8%. Compared with EP, the T_5%_ of EP/DPBSi composites is slightly reduced with the increasing content of DPBSi, attributed to the early decomposition of phosphor-phenanthrene and methoxy groups [14]. Besides, the addition of DPBSi significantly reduces the temperature of the maximum decomposition rate (T_max_) and the maximum decomposition rate (R_max_) of EP composites. These results suggest that DPBSi promotes the formation of char layers, which hinder and slow down heat transfer. According to previous studies, the increase in residual mass reflects the improvement in flame-retardant activity in the material in the condensed phase [38]. Herein, the char yield of EP/DPBSi composites increases—EP/DPBSi-2, EP/DPBSi-4, and EP/DPBSi-6 increase from 13.17% to 19.71%, 24.8%, and 28.17%, respectively.

The thermal degradation behavior of EP and EP/DPBSi composites in the air atmosphere is different from that in the N_2_ atmosphere. The relevant results are shown in Appendix A. Both EP and EP/DPBSi composites exhibit two thermal decomposition peaks. Because of oxidation, the initial decomposition temperature of the samples is lower than that of the N_2_ atmosphere [27,39]. Moreover, the introduction of DPBSi does not reduce the thermal degradation temperature and even helps improve the char yield, attributed to the low phosphorus content and the physical interaction between DPBSi and EPs. The T_max_ values of EP/DPBSi-4 are 293.2 °C and 545.2 °C, which are higher than those of EPs (287.1 °C and 534.1 °C, respectively).

The TGA results in the air and N_2_ atmospheres indicate that the char yield of EPs is lower than that of EP/DPBSi composites. In other words, DPBSi helps EPs retain more organic products during the charring process and reduce organic matter release.

### 3.3. Thermomechanical and Mechanical Properties

The dynamic thermal and mechanical properties of EP composites were evaluated by the dynamic thermomechanical analysis (DMA) test. Figure 5a shows the changeable curves of tan δ and storage modulus (E′) with temperature for EP and EP/DPBSi composites. Usually, the peak corresponding temperature is considered as the glass transition temperature (T_g_) in the curve of tan δ [40]. Table 3 lists the relevant performance parameters containing crosslinking density (Ve) and storage modulus (E′). The introduction of DPBSi improves the storage modulus and the peak value of tan δ simultaneously, illustrating that the strength and toughness of EPs are improved [31,41]. Besides, all curves of tan δ show a single peak, reflecting the good compatibility between DPBSi and the epoxy resin matrix [42]. The flexible siloxane structure of DPBSi is uniformly dispersed in epoxy resin, which can help disperse and absorb stress. With the increasing content of DPBSi, the T_g_ value of cured EP first increases from 146.3 °C to 158.4 °C and then decreases to 151.4 °C and 149.5 °C. The crosslinking density (Ve) of EP composites is higher than that of EPs, while EP/DPBSi-6 has a slightly lower crosslinking density (2.35 × 10^−3^ mol cm^−3^) than EP/DPBSi-4 (2.48 × 10^−3^ mol cm^−3^). These results can be explained by the following fact. On the one hand, it may be related to the benzimidazole groups. The lone pair electrons on the nitrogen atom in benzimidazole facilitate the curing degree of EP and generate physical cross-linking sites (e.g., hydrogen bonds and π–π interactions) in the polymer network, enhancing the dynamic thermo-mechanical properties of EPs [43]. On the other hand, with an increase in DPBSi content, the increased steric hindrance of DPBSi limits the positive role of benzimidazole groups, resulting in a slight decrease in T_g_ and Ve.

To further evaluate the effect of DPBSi on mechanical behavior, three-point bending, depth-sensing indentation and impact tests were conducted. The results of flexural strength, flexural modulus, impact strength, Young’s modulus, and hardness are shown in Figure 5b–f. Based on the experimental results (see Table 3 and Figure 5b), the flexural modulus and flexural strength of EP/DPBSi-6 are as high as 3205.4 MPa and 142.2 MPa, respectively, which are increased by 23.6% and 61.4%, respectively, relative to EPs. Meanwhile, compared with the unmodified EP, the impact strengths of EP/DPBSi-2, EP/DPBSi-4, and EP/DPBSi-6 are increased by 148%, 189%, and 240%, respectively. Such a large increase in impact strength is the result of the combination of Si-O bonds and plenty of rigid aromatic rings [44,45]. The flexible phenylsiloxane structure enhances the matrix toughness, thereby avoiding the brittle problem caused by many rigid groups. Moreover, as shown in Figure 5d,e, the Young’s modulus and hardness of EP/DPBSi-4 are increased by 9.3% and 19.1%, respectively. These results of depth-sensing indentation (DSI) tests suggest that DPBSi plays a role not only in toughening but also in reinforcing EP systems.

The morphologies of fracture surfaces after quenching with liquid nitrogen of EP samples are shown in Figure 6. EPs show a uniform and regular cross section, which is quite smooth and has little cracks. This means that the cross-sectional view of EPs belongs to the typical brittle fracture morphology. In contrast, with the increase in DPBSi content, the cross-sectional morphologies of EP composites become rough and show more cracks, showing a ductile fracture morphology [46]. It is worth mentioning that the second phase does not appear, indicating that DPBSi is uniformly dispersed in the EP matrix [32].

In summary, the remarkable improvement in mechanical properties benefits from the unique molecular structure of DPBSi, including flexible bonds (Si-O), rigid aromatic rings, and promoting groups (benzimidazole). More importantly, the good dispersibility of DPBSi enables these functional groups to play their roles effectively. The benzimidazole groups of DPBSi provide more cross-linking sites to enhance the interaction between the flame retardants and the EP network. Meanwhile, flexible Si-O bonds increase the impact resistance of the matrix and plentiful rigid aromatic rings help dissipate stress. This interesting design of molecular structure simultaneously enhances EP toughness and rigidity.

### 3.4. Flame Retardancy

The limiting oxygen index (LOI) and UL-94 vertical burning test are trenchant means for evaluating the flame-retardant behavior of EPs and EP/DPBSi composites. The related results are listed in Table 4 and the combustion process of UL-94 vertical combustion is exhibited in Figure 7. EP, a combustible material with a low LOI value of 23.6%, is accompanied by severe dripping during combustion. After the addition of DPBSi, the dripping disappears. The LOI value of EP/DPBSi-2 is promoted to 28.9% and achieves the UL-94 V-1 rating. Up to the point where the addition of DPBSi is 4%, the composite containing only 0.22% phosphorus can pass a V0 rating and the LOI value increases to 31.5%. Besides, EP/DPBSi-6 shows the best flame retardancy, with the highest LOI value of 33.5% and the shortest extinguished time in all samples. Because of the efficient catalytic carbonization and blowing-out effect, EP/DPBSi composites have better flame-retardant properties than EPs.

The cone calorimeter test (CCT) is widely used to simulate the burning behavior of materials under real fire. Some essential parameters are obtained to evaluate the fire safety of epoxy thermosets, including time to ignition (TTI), total smoke production (TSP), smoke production rate (SPR), peak of smoke production rate (PSPR), total heat release (THR), heat release rate (HRR), peak of heat release rate (PHRR), average effective heat of combustion (av-EHC), average CO yield (av-COY), fire growth rate (FIGRA), and residual morphologies. All the data are listed in Figure 8, Appendix A, and Table 5. From Appendix A, as the DPBSi content increases, more char residue remains and the char layer structure becomes more obvious.

Smoke is a secondary risk in fires and brings great hazards to humans. Compared with EP, TSP and PSPR of EP/DPBSi-6 are decreased by about 37% and 25%, respectively (see Figure 8a,b). Moreover, the main component of smoke is aromatic compounds, so the reduced release of smoke represents an increase in the amount of char residue, which is consistent with the results of TGA. For pure EP, THR is 89.5 MJ/m^2^ and PHRR is 1002 kW/m^2^. From Table 5, it is found that both THR and PHRR of EP/DPBSi composites display a noticeable decrease. Compared with EP, THR and PHRR of EP/DPBSi-6 are decreased by 43.4% and 32.5%, respectively. The reduction in THR and PHRR conforms to the results in LOI and UL-94 tests. In other words, DPBSi can weaken the combustion intensity of EPs.

Table 5 also lists av-COY and av-EHC. The trend of av-EHC and av-COY first decreases and then increases, attributed to insufficient combustion of volatile matter [20]. Changing trends suggest that the flame-retardant effect of DPBSi is more pronounced in the condensed phase than in the gas phase with the increase in DPBSi content. The gradual decrease in FIGRA reflects the contribution of DPBSi to improving the fire safety of EP [28].

### 3.5. Char Analysis

The microscopic morphologies of the char residue are analyzed by scanning electron microscopy (SEM) and exhibited in Figure 9. As shown in Figure 9a,e, owing to the complete burning of EP and severe melting droplets, the char layer of EP becomes smooth inside and loose outside. With the increase in DPBSi content, the char layer structure of EP composites changes significantly. As shown in Figure 9d,h, the inner char layer shows a distinct “honeycomb” structure and the outer char layer becomes dense for EP/DPBSi-6. The changes can be explained by the following fact. Firstly, when EP/DPBSi composites burn, the dense exterior char layer forms on the surface of the matrix under the catalysis of P/N/Si elements. Secondly, the dense exterior char layer causes the gas generated in combustion to escape slowly, eventually leading to the expansion of the inner char layer and the appearance of alveolate pores [36]. It is worth mentioning that these morphologies are more prominent with a higher DPBSi content, conforming to the TGA and CCT results. Because of this inner porous and outer dense char layer structure, the transmission of oxygen and heat is effectively insulated.

Raman spectra were utilized to evaluate the structural order of char residue. Two characteristic peaks around 1360 cm^−1^ and 1590 cm^−1^ correspond to disordered carbon (D band) and ordered carbon (G band), respectively. The D peak and the G peak are attributed to the A_1g_ vibration of disordered carbon and the E_2g_ vibration of ordered carbon, respectively [47]. The graphitization degree of the char residue is usually expressed by the integral ratio between the D band and the G band (I_D_/I_G_) [48]. The values of I_D_/I_G_ for EP/DPBSi-2, EP/DPBSi-4, and EP/DPBSi-6 were calculated to be 2.24, 2.01, and 1.94, respectively, all lower than EP (I_D_/I_G_ = 2.44). This signifies that EP/DPBSi-6 has the highest degree of graphitization among all EP samples, and the orderliness of the char layer is improved. From these results, it is concluded that DPBSi contributes to forming a compact char layer after the ignition of epoxy resins, which helps isolate the matrix and flammable areas.

After exploring the structural ordering of the char layer, the chemical composition of the char residue was analyzed by X-ray photoelectron spectroscopy (XPS). Because of the marked flame retardancy of EP/DPBSi-6, EP/DPBSi-6 was analyzed and the corresponding XPS spectrum is shown in Appendix A. High-resolution XPS spectra of preventive elements are revealed in Figure 10. For the C 1 s spectrum, the peak at 284.8 eV is the C-H and C-C contributions in aromatic compounds, indicating that the presence of a graphite-type structure. The peak at 286.1 eV is related to the bonds of C-O-P and C-N. In the case of Si 2p, two peaks are detected at 102.1 eV and 103.0 eV, which belong to the bonds of Si-C and Si-O/SiO_2_, respectively. These Si-containing structures are beneficial to consolidate the stability of char layers [49]. The N 1 s spectrum can be divided into four peaks at 398.8 eV (Pyridinic N), 399.7 eV (Amine), 400.4 eV (Pyrrolic N), and 401.1 eV (Graphite N), because benzimidazole groups are retained in the char layer by thermal carbonization [45]. The P 2p spectrum can be split into three peaks, which are P=O (133.5 eV), P-O-C (135.4 eV), and P_2_O_5_ (135.2 eV) [50]. This result illustrates that DPBSi facilitates the formation of the phosphoric acid-containing char layer, which elevates the flame retardancy in the condensed phase.

Subsequently, Fourier transform infrared (FTIR) spectroscopy is used to further determine the chemical structure of the char residue. The FTIR spectrum is shown in Figure 11. Compared with EP, EP/DPBSi-6 has new characteristic peaks at 1036 cm^−1^, 1232 cm^−1^, and 1361 cm^−1^, which are attributed to P−O−C, P=O, and Si−C, respectively. Besides, the peaks of Si-O and SiO_2_ appear separately at 1178 cm^−1^ and 1107 cm^−1^, respectively. It is worth mentioning that SiO_2_ is considered to help increase the stability of the char layers [51]. The formation of char layers being rich in aromatic structures during combustion can be illustrated by the enhanced absorption peaks at 1610 cm^−1^, 1510 cm^−1^, 822 cm^−1^, and 752 cm^−1^. It is worth mentioning that the signal at 1510 cm^−1^ is obviously stronger than the other three signals, attributed to the fact that the signal of C=N overlaps with the signal of the aromatic structure here. In addition, the peak at 1455 cm^−1^ belongs to C-N. It is illustrated that benzimidazole structure plays a role in the condensed phase. These abovementioned signal peaks demonstrate that the char layers are composed of highly carbonized aromatic networks catalyzed by benzimidazole groups, phosphoric acid compounds, and organic/inorganic silicon compounds.

### 3.6. Gaseous Products’ Analysis

The thermal pyrolysis products of EP and EP/DPBSi-6 were investigated by TG-FITR. The absorbance curves of EP and EP/DPBSi-6 over time at different temperatures are shown in Figure 12a,b. TG-FTIR 3D images of EP and EP/DPBSi-6 composites are shown in Appendix A. These images are similar in shape but markedly different in intensity. These situations can be explained by the fact that DPBSi contributes to reducing the number of gaseous products effectively. Meanwhile, EP and EP/DPBSi-6 show similar characteristic peaks, which appear at 3500–3700 cm^−1^ (H_2_O), 2970 cm^−1^ (hydrocarbons), 2400 cm^−1^ (CO_2_), 1740 cm^−1^ (carbonyl compounds), 1510 cm^−1^ (aromatic compounds), and 1258 cm^−1^ (ether compounds) [10,52]. Additionally, the increased peak height at 747 cm^−1^ (P-O-Ph) and 1260 cm^−1^ (P=O) is contributed by the DOPO groups, demonstrating that phosphorus compounds achieve the flame-retardant influence in the gas phase. To further compare the difference of gas products between EP and EP/DPBSi-6 intuitively, the time-dependent changes in intensity are summarized in Figure 12c–h. In the Gram–Schmidt curve shown in Figure 12c, it is observed that the total release of organic volatiles is decreased, meaning more organic products remain in the char residue. This is consistent with the previous TGA. Besides, as summarized in Figure 12d–h, the simultaneous reduction in the release of the representative organic volatiles also occurs. Therefore, with the aid of DPBSi, the thermal degradation of the matrix is inhibited and the smoke emission is reduced during the burning process.

### 3.7. Flame-Retardant Mechanism

Based on the above analysis, the flame-retardant mechanism of EP/DPBSi composites is exhibited in Figure 2. There is a synergistic flame-retardant effect between DOPO, benzimidazole, and phenylsiloxane groups in DPBSi. On the one hand, DOPO groups decompose quickly to release P-containing free radicals (e.g., HPO•, PO•, and PO_2_•), and the imidazole groups decompose to produce N-containing nonflammable gases. P-containing free radicals are used as sweepers for free radicals such as H• and OH•, while noncombustible gases can expand the char layers and dilute the concentration of organic volatile products in the gas phase. These actions interrupt the combustion reaction so that more organic products remain. On the other hand, the aromatic structure of DPBSi provides an adequate carbon source for the formation of char layers. Under the catalysis of phosphoric acid and phenylsiloxane groups, the char layers form quickly. Besides, benzimidazole groups can promote further cross-linking of char layers. The formed dense and porous char layers exert an effective barrier effect and prevent the continued combustion of the matrix.

This work is compared to the reported modified epoxy resins, which have excellent mechanical properties and flame retardancy. Table 6 lists relevant representative data. DPBSi, as a multifunctional flame retardant, exhibits similar or better effects in improving the flame retardancy and mechanical properties of EP.

## 4. Conclusions

To summarize, a facile strategy of combining vanillin with rigid–flexible groups, promoting groups and polar phosphorus groups, can effectively solve the high flammability and poor impact resistance of EPs. The presence of flexible groups (Si-O) and promoting groups (benzimidazoles) helps to form a rigid–flexible epoxy network. Compared with pure EP, the storage modulus, flexural strength, and impact strength of EP/DPBSi-6 are increased by 61.1%, 61.4%, and 240%, respectively. Because of the synergistic effect between P/N/Si elements, EP/DPBSi-4 with only 0.22 wt% phosphorus content shows a gratifying LOI value of 31.5% and passes the UL-94 V0 grade. Furthermore, PHRR and TSP of EP/DPBSi-6 are decreased by 32.5% and 37%, respectively. Such a flexible strategy enables the wider application of EPs and provides new ideas for constructing vanillin-based EP composites with excellent mechanical properties, efficient flame retardancy, and smoke suppression performance.

## Data Availability

The data supporting the findings described in this manuscript are available from the corresponding authors upon request.

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
