# Peer review of "A Novel P/N/Si-Containing Vanillin-Based Compound for a Flame-Retardant, Tough Yet Strong Epoxy Thermoset"

_polymers, 2023, doi:10.3390/polym15102384_

Round 1

Reviewer 1 Report

Article A Novel P/N/Si-Containing Vanillin-Based Compound for Flame-retardant, Tough yet Strong Epoxy Thermoset is interesting and relatively well written. However, before further editorial stages, the Authors must make a few corrections. My comments are listed below:

1.       The formulations of ingredients for the preparation of epoxy thermosets should also be given numerically.

2.       Authors presented in section 2.4 how to obtain resins. However, they did not disclose the size of the samples they obtained.

3.       Test equipment data in section 2.5 must be completed in accordance with the requirements of the Polymers journal.

4.       Did the authors also try to measure the obtained samples on a cone calorimeter with a different heat flux? Testing at 50 kW/m2 is more similar to real fire conditions. Did the authors also try to test samples with a thickness higher than 3 mm? I know from experience that 3 mm is at the lower limit of the thickness of the sample. Optimally, thin samples should be at least 5 mm, because this is important for obtained measurement data.

5.       Authors did not provide standard deviations from the results obtained on CC. Does this mean that the Authors examined only one sample of each? It is recommended that there should be a minimum of 3-5 repetitions of one sample, because even significant differences may occur between individual samples at small thicknesses.

6.       Figure 12 is not very legible. I suggest enlarging it.

7.       The Conclusion section should be more supported by research results as required by the Polymers journal.

The English language of the article requires minor linguistic corrections.

Reviewer 2 Report

The authors have been submitted a comprehensive study on a novel and effective vanillin-based flame retardant titled „A Novel P/N/Si-Containing Vanillin-Based Compound for Flame-retardant, Tough yet Strong Epoxy Thermoset”. Congratulations for the nice results.

I can recommend publication of this article in the journal Polymer conditional on addressing the relevant points below. The subject of the paper is very important, and the manuscript falls within the scope of the Journal.

The following items should be addressed:

I recommend rewriting the abstract with less data. I am sure these data are important, however, in an abstract less is better.

Please define the abbreviations first when it used. (For example THR, TSP and pHRR). Maybe many of them trivial for the authors and specialists, but not for every polymer scientist. Also, the definition of DGEBA is missing. Please define all of them.

In introduction, please mention a few examples about the existing P-N containing flame retardant additives applied for EP. For example, Journal of Macromolecular Science-Chemistry 1986, A23, (1), 19-36.; Polymer Degradation and Stability 2006, 91, (3), 585-592.

Aging is a relevant phenomenon in the field of materials sciences. Please indicate how long after the synthesis of the polymer the measurements were taken, and whether any changes were observed as a function of time.

In Experimental Section, the name of the compounds were written with initial capital letters, they have to be corrected to uncapital ones. Please use spaces between numbers and units.

In row 100, “Reaction complete” (instead of enough)

In Experimental, chapter 2.3 and 2.4, please write the exact amounts. Only wt% was mentioned in Table 1.

In the case of characterization, 13C NMR of the additive is missing. I highly recommend to add this also. If the NMR sample have been prepared once, there is no reason not to measure it. I also suggest to add the curde data (like NMR fid files) as supplementary files also. Maybe it is also possible.

In scheme 2, fused N-heterocyclic moieties were drown into the structure of the char. Was the presence and binded form of nitrogen proved by any analytical method? (Aromatic-non aromatic, etc.) (Fig S4?)

Please add the meaning of non-trivial abbreviations into the figure/table caption also.

Please add the DOI to the references.

In row 446, please remove “Please add:”

Round 2

Reviewer 1 Report

Article can be published in present form.